# Image Classification Using Multiple Convolutional Neural Networks on the Fashion-MNIST Dataset

**DOI:** 10.3390/s22239544

**Published:** 2022-12-06

**Authors:** Olivia Nocentini, Jaeseok Kim, Muhammad Zain Bashir, Filippo Cavallo

**Affiliations:** 1Department of Industrial Engineering, University of Florence, 50139 Florence, Italy; 2The BioRobotics Institute, Sant’Anna School of Advanced Studies, 56127 Pisa, Italy

**Keywords:** image classification, convolutional neural networks, dressing assistance, social robotics

## Abstract

As the elderly population grows, there is a need for caregivers, which may become unsustainable for society. In this situation, the demand for automated help increases. One of the solutions is service robotics, in which robots have automation and show significant promise in working with people. In particular, household settings and aged people’s homes will need these robots to perform daily activities. Clothing manipulation is a daily activity and represents a challenging area for a robot. The detection and classification are key points for the manipulation of clothes. For this reason, in this paper, we proposed to study fashion image classification with four different neural network models to improve apparel image classification accuracy on the Fashion-MNIST dataset. The network models are tested with the highest accuracy with a Fashion-Product dataset and a customized dataset. The results show that one of our models, the Multiple Convolutional Neural Network including 15 convolutional layers (MCNN15), boosted the state of art accuracy, and it obtained a classification accuracy of 94.04% on the Fashion-MNIST dataset with respect to the literature. Moreover, MCNN15, with the Fashion-Product dataset and the household dataset, obtained 60% and 40% accuracy, respectively.

## 1. Introduction

Caretakers are needed as the elderly population rises, which may put society in an unsustainable situation. This condition increases the need for automated assistance and social robots will be especially useful for carrying out daily tasks [1]. Everyday tasks such as manipulating clothing are a challenge for robots. Key components of clothing manipulation include detection and classification.

Classifying objects by looking at them is a trivial task for humans but challenging for computers. To perform the task on a par with humans, the computers must also be robust to carry out the recognition under varying light conditions, different sizes and colors of objects, occlusions, etc. Despite the difficulty in achieving classification ability, a huge amount of literature has developed on various methods to classify objects. As a result, it is becoming a basic ability of any intelligent agent and has wide-reaching applications in robotics. The huge amount of literature explains it with various methods attempting to classify objects with greater accuracy.

Early attempts at solving the classification problem involved meticulously defining and extracting certain features from image datasets, so those characteristics represented most of the data with high confidence. These features were defined so that they aimed to capture interesting information in images such as edges, circles, lines, or a combination of these, which are ideally invariant to translation, scale, and varying light intensities. HOG [2], SURF [3], SIFT [4], and FAST [5] are a few of them. Once these features were extracted, classifiers such as Support Vector Machine [6], Naive Bayes [7], Decision Trees [8], K-Nearest Neighbors [9] or Linear Discriminant Analysis [10] were employed to decide the membership of an unseen image. These methods, however, were time-consuming and defining features, and capturing a wide range of information, was difficult. Consequently, recent works have given way to learn these features using Convolutional Neural Networks (CNNs) instead of hand-crafting them.

CNN, an extension class of Artificial Neural Networks (ANNs [11]), is a class of supervised learning methods whereby huge amounts of data are fed into them. Moreover, these networks learn several convolution filters, which are capable of detecting interesting features in the dataset by minimizing a certain loss function. This approach has been extremely successful in solving object classification problems. This has opened up new avenues for such methods being used in a range of applications of which clothing classification is one of them.

There have been various attempts at sorting fashion images into different categories using CNNs. These attempts differ from one another based on the network’s structure, such as the number and size of convolutional layers, adding residual blocks [12] or adding Long Short-Term Memory (LSTM) units [13], etc. Based on the improved models over the year using CNNs, we investigated a new network’s structure which uses multi-convolutional layers system for fashion image classification. From the state of art, it can be pointed out that using CNNs boosts the accuracy of clothing image classification; therefore, we decided to study in depth a specific case of CNNs, the multiple convolutional layers (MCNNs), since these models have fewer parameters than a traditional CNN (e.g., params of Alexnet: 58, 302, 196, params of Resnet18: 11, 175, 370), which make the networks more efficient [14].

The main contributions of this work are the following:We proposed new MCNN models to increase the classification performance on the Fashion-MNIST dataset. Moreover, searching hyper-optimization and data augmentation techniques are applied to improve the generalization of the models.We created the Fashion-Product dataset and a customized dataset of ours for confirming the network´s performance.We compared our models’ performance with state-of-the-art and the literature of different model structures trained on the Fashion-MNIST dataset. In addition, the performance on the Fashion-Product dataset and a customized dataset are compared by state-of-the-art and MCNN15.

The paper is composed as follows: Section 2 presents the related works. Section 3 describes the methodology related to fashion image classification; Section 4 introduces the description of the experimental set-up, and Section 5 contains the results. Section 6 discusses the experiments. Finally, Section 7 concludes the paper.

## 2. Related Works

Many datasets are used for clothing image classification, such as the Fashion-MNIST, the DeepFashion-C, the AG dataset, and the IndoFashion dataset. Concerning the DeepFahion-C, this dataset was used in [15]. In this work, the authors suggested a framework for retrieving fashion products based on images that draws inspiration from biology and resembles the two-stream visual processing system the human brain is thought to have. The attentional heterogeneous bilinear network (AHBN) is hypothesized to include a fully convolutional branch and a deep CNN branch; both are used to extract landmark localization data and fine-grained appearance attribute information, respectively. Following the application of a compact bilinear pooling layer to simulate the interaction of the two streams, a combined channel-wise attention mechanism is then used to focus on significant channels in the derived heterogeneous features. The DeepFashion-C was also used in [16]. Their fashion model integrates two attention pipelines, landmark-driven and spatial-channel attention, to improve apparel classification. Through the use of these attention pipelines, their model was able to represent the multiscale contextual information of landmarks, which enhances the accuracy of classification by determining the locations of the most crucial features in an input image. In [17], the authors introduced a semi-supervised multi-task learning strategy to achieve attribute prediction and apparel category classification. They adopted a teacher–student (T-S) pair model that uses weighted loss minimization while exchanging knowledge between teacher and student to intensify semi-supervised fashion clothing analysis. With the simultaneous learning of labeled and unlabeled samples, they aimed in this study to increase the feature representation and prevent further training for unlabeled examples. The authors evaluated the proposed approach on the large-scale DeepFashion-C dataset and the combined unlabeled dataset obtained from six publicly available datasets. The AG dataset was used instead in [18], where a real-world study was proposed, which was aimed at automatically recognizing and classifying logos, stripes, colors, and other features of clothing, solely from final rendering images of their products. The IndoFashion was introduced in [19], a dataset of over 106,000 images with 15 different categories for the fine-grained classification of Indian ethnic clothes.

The apparel classification task with the Fashion-MNIST dataset has been developed in several works of literature, and the accuracy obtained is, in some cases, more than 90%. CNNs are widely used for clothes classification with this dataset, and two examples of that were presented in [20,21]. In the first work, the authors implemented a CNN by retraining the final layer of a GoogLeNet to classify apparel. In the second one, the Fashion-MNIST dataset was also tested on two networks created by the authors (CNN-Softmax and CNN-SVM). In [22], the authors proposed three different CNN architectures and used batch normalization and residual skip connections to ease and accelerate the learning process. Other CNN architectures were applied in [23]. The authors presented four different CNN models for training the Fashion-MNIST dataset, comparing their method with state of the art. Additionally, in [24], the authors proposed to apply Hierarchical Convolutional Neural Networks (H-CNN) to apparel classification. This study has contributed to being the first trial to apply the hierarchical classification of apparel using CNN. The authors implemented an H-CNN using VGGNet on the Fashion-MNIST dataset, obtaining a decreasing loss and an improved accuracy compared to the literature.

In [25], other networks such as ResNet, a Wide Residual Network (WRN), and a PyramidNet were used on the Fashion-MNIST dataset for image classification tasks. The authors improved the performance level by increasing the network width and number of channels. In addition, data enhancement and learning strategies had a greater impact on the model performance. In [26], the authors used a VGG-11 network to classify the same dataset, and they modified the network by introducing a multi-nonlinearity layer. This layer increased the learning of more complex patterns at a relatively small cost. Simultaneously, the batch normalization layer was added after each pooling layer, making it easier to train effective models by standardizing input data to make the distribution of each feature similar. In [27], the authors used a batch normalization strategy with a novel shallow convolutional neural network (SCNNB) to improve image classification accuracy. Moreover, in [28], the authors stated that with normalization, some tuning, and reduction of overfitting, they obtained an accuracy of 91.78% with a VGG-like architecture. They also tested a CNN with the Fashion-MNIST dataset and attained almost the same test accuracy of 90.77%. The authors pointed out that VGG produces better results but at the cost of taking a long time to train and being more computationally intensive.

As for concerning apparel classification using Support Vector Machine (SVM) and Fashion-MNIST dataset, in [29], the authors proposed fashion articles classification system using HOG feature space and multiclass SVM classifier, obtaining an accuracy of 86.53%. Additionally, in [30], the authors compared the performance of different models (SVM, K-Nearest Neighbors, Random Forest, Decision Tree and a CNN) on the Fashion-MNIST and CIFAR-10 datasets. Their work also examined different feature extraction techniques to improve the model’s performance. From the results, it was shown that the approach of using an autoencoder was better than the Principal Component Analysis (PCA) for boosting the performance of the model; especially, using an autoencoder with SVM surpassed the performance of a CNN model.

A different network called LSTM was used in [31] to boost the model’s accuracy in the clothes classification task using the Fashion-MNIST dataset. In [32], an LSTM architecture was also used for image classification on the Fashion-MNIST Dataset. They used cross-validation that detected and prevented overfitting, and fine-tuning helped to improve the model’s structure, increase the score and reduce training time consumption. Moreover, Heuristic Pattern Reduction methods reduced the training time. In some cases, they could also increase the score, while network pruning was one of the most significant challenges in the experiment.

Hyperparameter optimization and regularization techniques, such as image augmentation and dropout, are used to improve the accuracy of networks in classification tasks using the Fashion-MNIST dataset. In [33], this technique was used with four-layer ConvNets, which could attain an accuracy of 93.99%. From the literature, lots of methods for training the Fashion-MNIST dataset were used and demonstrated great performances during testing; despite this, there is still room for improvement concerning the accuracy of this kind of dataset.

## 3. Methods

### 3.1. Multiple Convolutional Neural Networks Models Design

Convolutional neural networks (CNNs) have a new paradigm for computer vision [34]. Furthermore, it is broadly used for visual recognition and succeeded in identifying many objects during the Imagenet classification Challenge [35]. However, this network still needed to catch up to human object recognition performance (95% of success). In 2015, ResNet [12] dramatically increased object recognition performance using a residual connection. Moreover, better performance with the small network model [36] and model scaling method (width, depth and resolution) were proposed [37]. Inspired by these different architectures, a multiple CNN structure has the potential to build a new network that improves the performance of classification [38]. We chose to use this kind of network to boost the accuracy of clothing image classification, since these models have fewer parameters than a traditional CNN, which make the networks more efficient [14].

Our proposed MCNN model structure took inspiration from [14], which is a deeper network improving the extraction of the spatial features on the Fashion-MNIST dataset. This new MCNN model consists of three convolutional layer groups, three max-pooling layer groups, one fully connected layer and a final softmax layer (Figure 1). The convolutional layer groups of our MCNN are composed of a kernel size 3 × 3 (fixed), one padding, and the number of the input and out channels in each convolutional layer is selected by the Ray tune [39] library, which is explained in the next section. Moreover, batch normalization and RELU activation functions are applied after each convolutional layer, and a max-pooling layer is inserted after each convolutional layer group. Then, a fully connected layer is included before softmax layers that flatten 2D spatial maps for image classification. Originally, the result of our MCNN occurred overfitting, so we applied the regularization term with optimization, and the cross-entropy is chosen as follows:(1)$$Loss=-\frac{1}{n}\sum_{i=1}^{n}\sum_{j=1}^{K}\left[y_{ij}\cdot log\left(p_{ij}\right)\right]$$
where *i* and *j* represent, respectively, index samples and classes. *N* is the number of samples, *K* is the number of classes, $y_{ij}$ is the label, and $p_{ij}$ is the softmax prediction. Comparing the performance of the MCNN model, a different number of convolutional layers, which included three (MCNN9), four (MCNN12), five (MCNN15), and six (MCNN18) convolutional layers in each convolutional layer group, are designed and tested (see Figure 1).

### 3.2. Hyperparameters Optimization

Many hyperparameter optimization methods improve the network performance when constructing neural networks. To select them properly, some hyperparameters tuning algorithms, such as HyperOpt [40], Optuna [41], Ray tune [39], PBT [42], and so on, have been developed; these algorithms support random search, Bayesian optimization, early stopping, etc. Between the hyperparameter tuning algorithms, Ray tune can be integrated with many optimization libraries. It can also provide scale and flexibility, and these characteristics brought us to use this algorithm with our model. We kept the entire structure of our models composed of three convolutional layer groups and a fully connected layer. However, we considered the number of input and output filters in each convolutional layer, the number of features in a fully connected layer, the batch size and the kernel size, which mainly affect the network’s performance. We used Adam optimizer with a fixed learning rate (0.001) based on [43,44]. In addition, graphic card performance considers the batch size, and the epoch sets a fixed value to confirm overfitting. Without a regularization term, our model has overfitting under epoch 50, so we added a regularization term and kept training our models until reaching epoch 100. Based on this process, we can investigate the testset’s loss whether overfitting or not. Last, we could not apply for different stride sizes because of the small input size and many convolutional layers (our model used stride = 1).

The hyperparameter configuration with different values is provided and set based on our GPU memory performance (see Table 1). It is randomly used to discover optimal hyperparameter values using Ray tune [39].

## 4. Experimental Set-Up

Our experimental set-up is designed with the help of a household object dataset collected by a Microsoft Kinect V2 mounted on the top of a tripod. We trained the Fashion-MNIST dataset, and we tested the Fashion-Product dataset and a customized dataset of ours on a PC running Ubuntu 16.04 LTS and using a GTX-1080ti GPU. Predominantly, the code was written in Python. For the network models training, we used the Pytorch library [45] with the Adam optimizer and tuned the hyperparameters (learning rate = 0.001, regularization term = $1\times 10^{-5}$, batch size = 64, and epoch = 100).

Data augmentation (random horizontal flip and random affine techniques from Pytorch) was applied to increase the model’s performance and prevent overfitting.

### 4.1. Datasets

We used the Fashion-MNIST dataset for training the models, and we tested the architectures with the Fashion Product and our customized dataset.

#### 4.1.1. Fashion-MNIST Dataset

The Fashion-MNIST dataset is a collection of Zalando’s fashion objects, having a training set of 60,000 examples and a test set of 10,000 examples. Each example is a 28 × 28 grayscale image. The dataset contains four files, including the labels and images, and each image is associated with a label from 10 classes, as shown in Table 2.

We chose this dataset for two reasons: the first one is that it is complex, and some classifiers only rarely can achieve a perfect score on it, so there is still an open window for optimization. The second reason is that some researchers used it for testing new methods, so that we can compare our results with many works.

#### 4.1.2. Fashion-Product Dataset

This dataset (Figure 2a) is composed of 44,441 images (https://www.kaggle.com/paramaggarwal/fashion-product-images-small, accessed on the 26 October 2022). We selected 10 images for the 10 categories of the Fashion-MNIST dataset for a total of 100 items. During the selection of the images, we decided to use most of the pictures not containing people wearing clothes, shoes, and bags. This choice was made since we reduced variables that created a dataset similar to the Fashion-MNIST dataset. To convert the RBG images to pictures similar to the Fashion-MNIST dataset, we applied the following steps: (1) resize the shortest side to 28 pixels, (2) center crop, (3) convert to grayscale with one channel, (4) normalize to [0–1], and (5) convert white background to black as in Fashion-MNIST.

#### 4.1.3. Customized Our Dataset

We collected our dataset (Figure 2b) at home for testing our models. We selected 50 items in our wardrobe, five for each of the 10 categories that are the same as the Fashion-MNIST dataset. With a Microsoft Kinect V2, we collected a picture of each object on the ground. To have a neutral background on where to put the objects, we laid a white blanket down on the ground.

## 5. Results

### Quantitative Results

In the results section, we compare four network models with the state of art architectures. Table 3 shows the accuracy related to each model. As it can be seen, the MCNN15 model obtained the highest accuracy percentage (94.04%) compared to the other architectures, while the lowest model was the one associated with Lenet (90.16%).

In Table 4, we compared our highest model (MCNN15) accuracy with other literature architectures. As can be observed from this table, our model obtained the highest accuracy (94.04%) compared to other architectures and had better performance than the lowest one that had an accuracy of 86.53% (SVM + HOG model).

In Figure 3a,b, the loss and the accuracy progress are manifested and, as can be seen, the accuracy of MCNN15 achieved 94.04%, while the loss was constantly decreasing and not overfitting.

Finally, the confusion matrix of the MCNN15 model with the Fashion-MNIST dataset is shown in Figure 4.

From the confusion matrix, it can be seen that the most accurate results were obtained from sneakers, bags, and trousers. Ankle boots were also well recognized, and shirts were more complicated to be recognized, since they were mostly confused with T-shirts/Tops.

The area under the receiver operating characteristic (ROC) curve for MCNN15 is 0.998 (see Figure 5); that described work has high performance. The area under the ROC curve (AUC) calculation summarized the ROC curve analysis into a scalar value, which ranges between 0 and 1. The nearer the AUC score to value 1, the better the application’s overall performance.

Concerning the Fashion-Product Dataset, we obtained the result using the state of art models with our model (see Table 5). The accuracy of each model is similar to the result in Table 3, but the maximum score of each category shows differently. The most recognized apparel categories by the models are marked in bold in Table 5. Comprehensively, MCNN15 is the best network model that could obtain a total of 60% accuracy.

Regarding our customized dataset, our model obtained 40% accuracy, as shown in Table 6. Based on Table 3, Mobilenet or Efficientnet should be ranked second, but VIT presented better performance than them. To understand the models’ performance for recognizing the categories, we marked scores in bold in Table 6. Overall, MCNN15 is also the best network model that could obtain a total of 40% accuracy.

We decided to test the models with these two datasets for the following reasons. First of all, the Fashion-Product dataset has more shapes compared to the Fashion-MNIST dataset, and secondly, it has different shape objects with the same category of the Fashion-MNIST dataset, so we wanted to understand how the networks behaved with this kind of dataset, which is different from the Fashion-MNIST dataset. We also chose to test the networks with a customized dataset of ours, since the Fashion-Product dataset is a dataset without “real” clothing images; in our dataset, wrinkles can be seen, and the shapes are softer and “real”.

## 6. Discussion

This section discusses the image classification of the Fashion-MNIST, the Fashion-Product dataset, and our dataset.

Concerning the first dataset, we obtained a higher accuracy (94.09%) compared to the state of the art. This could be related to the addition of a certain number of convolutional layers, as seen in our model (MCNN15). From the results section, it can be pointed out that adding new layers improves the model’s accuracy until the addition of 15 layers. After increasing this number of layers, the accuracy decreases (MCNN18), which may be caused by the quality of feature extraction from the MCNN18 model. It means that the Fashion-MNIST, a small image size dataset, could lose feature information over a certain number of convolutional layers and pooling operations [47] (our case is MCNN18). Moreover, the MCNN18 and bigger models need more training time without improving their performance.

In Table 4, the numbers in red represent the accuracy obtained using Pytorch. We realized that the model’s accuracy increased greatly when the authors trained their models using Tensorflow2. Therefore, we replicated the models and trained them using PyTorch again. As a result, accuracy is reduced to 90.64% compared to 98.8% on the CNN LeNet-5 model [23] and 90.25% compared to 99.1% on the CNNs model [23].

For what concerns Figure 3b, a regularization term is applied to keep the loss data stable and increase the generalization performance. However, when it is not applied, the accuracy increases by one or two percent with overfitting. In the confusion matrix, most fashions are satisfactorily classified. In particular, sandals, trousers, bags and sneakers are the classes that obtain a higher success rate, since they have specific shapes. However, when shirts are classified, our architecture confused them as T-shirts/Tops, which is why our model’s performance decreases.

From Table 5 and Table 6, it can be seen that our model could classify the unseen fashions. However, Table 5 shows that our model could not classify the shirt group and had a low success rate in the pullover, sneaker and bag classes. Furthermore, in Table 6, the model’s output could not classify coat, sandal, and sneaker classes despite the model being trained with data augmentation. It means that our model could not find similar examples of test images.

Concerning the network complexity in terms of the number of parameters, we investigated the trade-off between accuracy and network complexity in Table 7. As more convolutional layers are added, the parameter is dramatically increased. A deeper network model has more parameters, and it would perform better based on the deep learning theory. However, more parameters do not guarantee better performance. For example, Alexnet requires the most parameters (because of fully connected layers) in Table 7, but the performance is not the highest. In addition, although MCNN15 has fewer layers than MCNN18, MCNN15’s accuracy is higher than MCNN18. It would have a problem that a vanishing/exploding gradient would have occurred when the network model went too deep (Resnet solved this problem with the skip connection). In the future, we plan to investigate deeper network models with our structure to confirm this theory. Apart from this, few parameters (similar to MCNN15) with different network structures could show sufficient performance (e.g., Mobilenet).

One interesting point is that we expected our model could classify the most scores in each category, but as described in Table 5 and Table 6, each model recognized some categories better than our model. In addition, VIT shows worse performance in the Fashion-MNIST dataset but good performance in the customized dataset. For this reason, we could investigate different network models based on the results of the state-of-the-art models for future work.

It is a big challenge of supervisory learning that if we do not provide lots of good examples (similar examples) for creating a model during training, the model might fail to classify, even though it is in the same category. Our model still has the limitation of supervised learning, and GAN [48], or self-supervised learning [49] could be considered for overcoming the state-of-the-art problem.

## 7. Conclusions

Classifying household object images represents an issue in the object-based manipulation field. In this paper, we proposed a new model (MCNN15) with the highest accuracy (94.09%) compared to literature concerning image classification using the Fashion-MNIST dataset. We also evaluated our model with two other datasets (the Fashion-Product dataset and a household object datasets), even if we achieved an accuracy of 60% and 40%, respectively. Our proposed model does not dramatically increase the performance, but MCNN, which is not a new network structure, is still a promising network model that might generalize and improve the performance of the Fashion-MNIST dataset. Moreover, different hyperparameter optimization methods (different number of fully connected layers, batch size, stride, with or without dropout, etc.) could improve the model’s performance.

In the future, we would like to improve the accuracy of our model (MCNN15) with the Fashion-Product and household object datasets, changing the typology of layers and some parameters inside the layers. Additionally, we would like to implement a new model for fashion image classification combining some existing techniques [48,49] to boost the accuracy of the architecture.

## Figures and Tables

**Figure 1 sensors-22-09544-f001:**
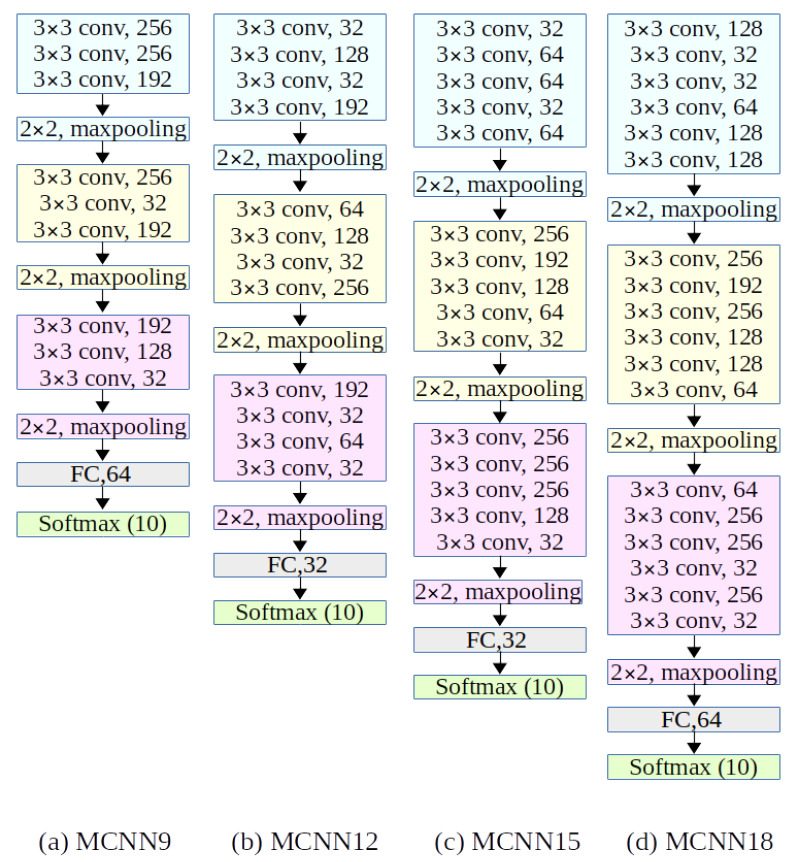
The architecture of the proposed multiple convolutional neural network models with different convolutional layers and optimized hyperparameters.

**Figure 2 sensors-22-09544-f002:**
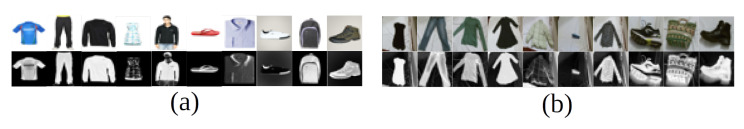
In (**a**) The Fashion Product Images Dataset (https://www.kaggle.com/paramaggarwal/fashion-product-images-small, accessed on the 26 October 2022) and in (**b**) the Fashion household dataset.

**Figure 3 sensors-22-09544-f003:**
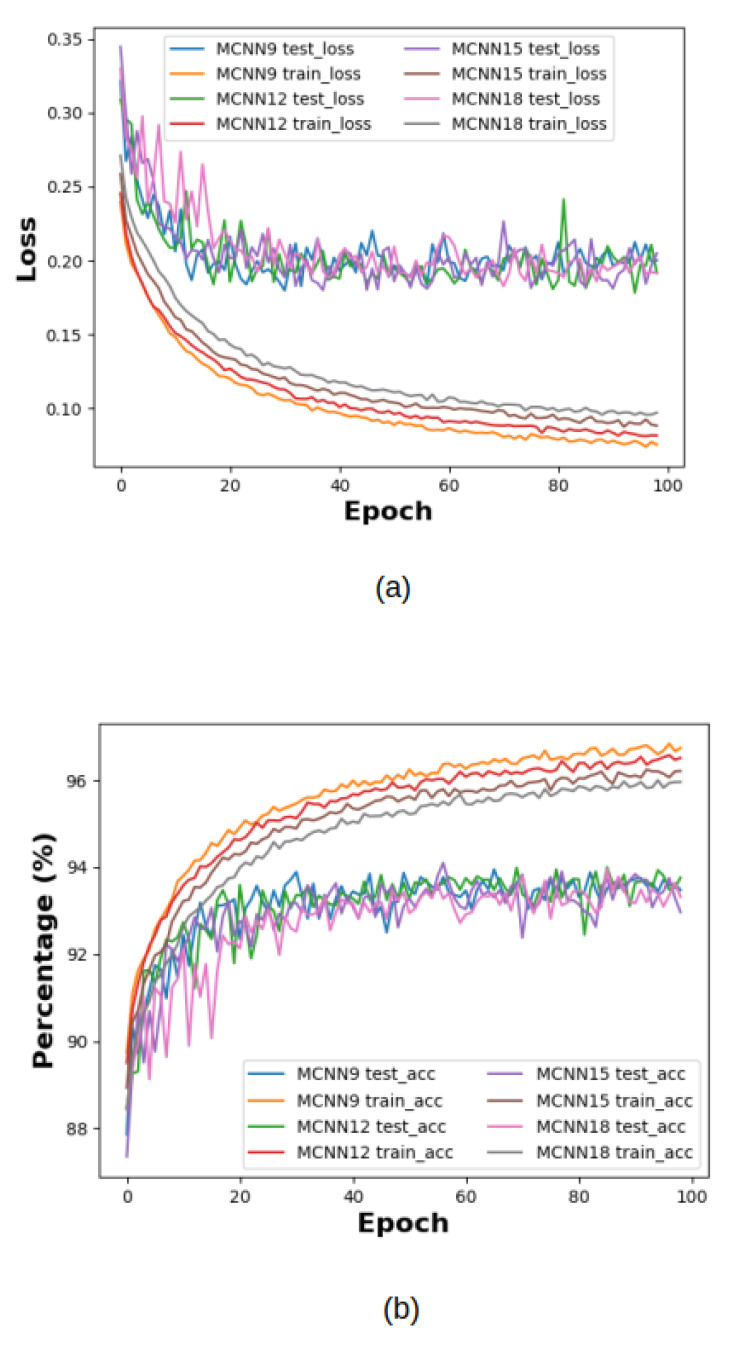
In (**a**), the progress of losses related to our model is shown, and in (**b**), the progress of accuracies is described.

**Figure 4 sensors-22-09544-f004:**
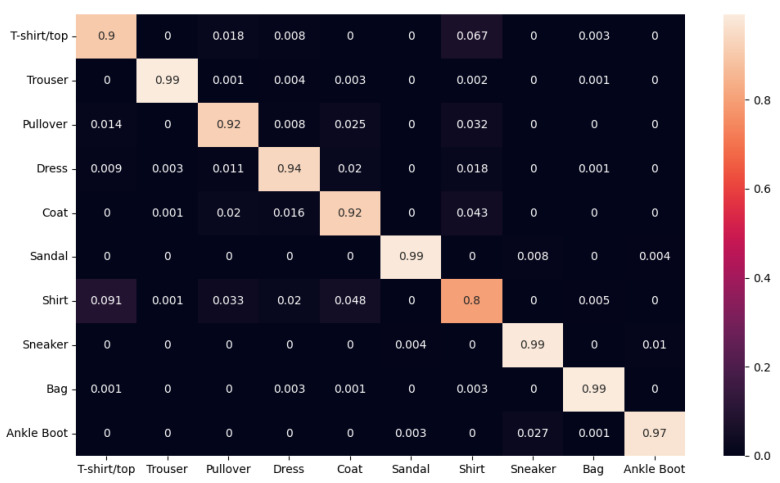
The confusion matrix of our MCNN15 model using the Fashion-MNIST dataset.

**Figure 5 sensors-22-09544-f005:**
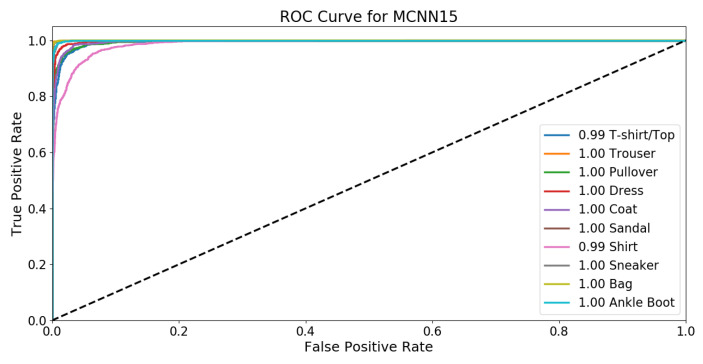
The plotted receiver operating characteristic (ROC) curve for MCNN15.

**Table 1 sensors-22-09544-t001:** Searching for optimal hyperparameter with values.

Parameter	Values
Batch size	(2, 4, 8, 16, 32, 64, 128, 256)
Kernel size	(1, 2, 3)
Number of filters	(32, 64, 128, 192, 256)
Fully connected layer	(32, 64, 128, 192, 256, 512, 1024)

**Table 2 sensors-22-09544-t002:** The Fashion-MNIST dataset is presented, which is composed of grayscale images divided into 10 classes.

Labels	Description	Examples
0	T-shirt/Top	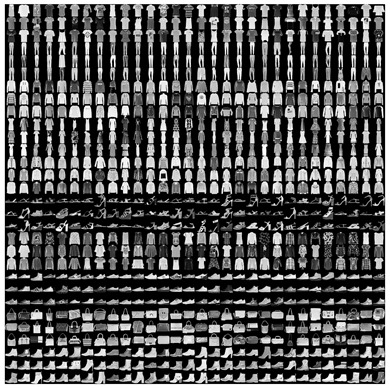
1	Trouser	
2	Pullover	
3	Dress	
4	Coat	
5	Sandal	
6	Shirt	
7	Sneaker	
8	Bag	
9	Ankle Boot	

**Table 3 sensors-22-09544-t003:** The performance of our models compared to the state of art. The bold format represents the result obtained with our models.

Model	Accuracy
Lenet [34]	90.16%
Alexnet [35]	92.74%
Resnet18 [12]	93.20%
Mobilenet [36]	93.96%
Efficientnet [37]	93.64%
VIT [46]	90.98%
**MCNN9**	**93.88%**
**MCNN12**	**93.90%**
**MCNN15**	**94.04%**
**MCNN18**	**93.74%**

**Table 4 sensors-22-09544-t004:** The performance of our model compared to literature architectures trained on the Fashion-MNIST dataset. The bold format represents the result obtained with our model. The red values represent the model trained with Pytorch.

Model	Accuracy	
H-CNN model with Vgg16 [24]	94%	
CNN-Softmax [21]	91.86%	
LSTMs [31]	88.26%	
LSTM [32]	89%	
CNNs [23]	90.25%	(99.1%)
CNN+HPO+Reg [33]	93.99%	
CNNs [22]	92.54%	
CNN LeNet-5 [23]	90.64%	(98.8%)
SVM+HOG [29]	86.53%	
CNN [30]	89.54%	
Vgg [28]	92.3%	
Shallow convolutional neural network [27]	93.69%	
VGG Network [26]	91.5%	
**MCNN15**	**94.04%**	

**Table 5 sensors-22-09544-t005:** The result of the Fashion-Product Images Dataset. The bold value represents the best result obtained for different networks.

	Lenet	Alexnet	Resnet18	Mobilenet	Efficientnet	VIT	MCNN15
T-shirt/top	9/10	**10/10**	7/10	9/10	9/10	**10/10**	**10/10**
Trouser	8/10	8/10	**10/10**	**10/10**	**10/10**	**10/10**	9/10
Pullover	2/10	3/10	4/10	**7/10**	3/10	1/10	4/10
Dress	3/10	4/10	3/10	3/10	1/10	1/10	**7/10**
Coat	**8/10**	7/10	**8/10**	**8/10**	7/10	**8/10**	**8/10**
Sandal	4/10	6/10	5/10	4/10	6/10	6/10	**7/10**
Shirt	0/10	0/10	0/10	0/10	0/10	0/10	0/10
Sneaker	1/10	3/10	**5/10**	4/10	0/10	0/10	3/10
Bag	8/10	8/10	7/10	**9/10**	**9/10**	8/10	4/10
Ankle boot	0/10	1/10	7/10	3/10	**9/10**	1/10	8/10
Total	43/100	50/100	56/100	57/100	54/100	45/100	**60/100**

**Table 6 sensors-22-09544-t006:** The result of dataset from household fashion. The bold value represents the best result obtained for different networks.

	Lenet	Alexnet	Resnet18	Mobilenet	Efficientnet	VIT	MCNN15
T-shirt/top	0/5	1/5	0/5	0/5	0/5	0/5	**3/5**
Trouser	0/5	4/5	0/5	1/5	2/5	**5/5**	4/5
Pullover	2/5	1/5	**4/5**	1/5	0/5	3/5	0/5
Dress	1/5	0/5	1/5	0/5	0/5	2/5	**3/5**
Coat	0/5	**1/5**	0/5	**1/5**	0/5	0/5	0/5
Sandal	0/5	0/5	0/5	0/5	0/5	0/5	0/5
Shirt	1/5	2/5	1/5	**3/5**	0/5	1/5	2/5
Sneaker	0/5	0/5	0/5	0/5	0/5	0/5	0/5
Bag	**5/5**	1/5	**5/5**	**5/5**	**5/5**	**5/5**	3/5
Ankle boot	0/5	0/5	4/5	0/5	3/5	0/5	**5/5**
Total	9/50	10/50	15/50	11/50	10/50	16/50	**20/50**

**Table 7 sensors-22-09544-t007:** The model network complexity in terms of accuracy and the number of parameters. The bold value represents the result of our model.

Model	Accuracy	Number of Parameters
Lenet [34]	90.16%	47,540
Alexnet [35]	92.74%	58,302,196
Resnet18 [12]	93.20%	11,175,370
Mobilenet [36]	93.96%	2,236,106
Efficientnet [37]	93.64%	4,019,782
VIT [46]	90.98%	212,490
**MCNN9**	**93.88%**	655,434
**MCNN12**	**93.90%**	971,882
**MCNN15**	**94.04%**	2,595,914
**MCNN18**	**93.74%**	3,004,362

## Data Availability

Not applicable.

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
