# Peer review of "Image Classification Using Multiple Convolutional Neural Networks on the Fashion-MNIST Dataset"

_sensors, 2022, doi:10.3390/s22239544_

Round 1
Reviewer 1 Report
This paper presented on the fashion image classification with four different neural network models. The authors used existing datasets to compare the performance of their proposed Multiple Convolutional Neural Networks with other deep learning models.
The content of this paper is not suitable for the Sensors journal. There is no information on the sensors used in the experiment. It is more on the exploration of different types of deep learning models to classify the fashion image. The main contribution of this paper is also not clear. It is lacking in term of the novelty and depth of the methodology proposed in this paper.
Author Response
Response to Reviewer 1 Comments
Dear Editors and Reviewers, thank you for your letter and for the reviewers’ comments concerning our manuscript entitled “Image Classification using Multiple Convolutional Neural Networks on the Fashion-MNIST dataset”. We have analysed these valuable comments thoroughly and have made corrections which we think fully address the reviewers’ concerns, as described below.
Reviewer 1
Point 1: The content of this paper is not suitable for the Sensors journal. There is no information on the sensors used in the experiment. It is more on the exploration of different types of deep learning models to classify the fashion image. The main contribution of this paper is also not clear. It is lacking in term of the novelty and depth of the methodology proposed in this paper.
Reply: We appreciate this comment. In the experimental set-up section we used a Kinect V2 (see line 197 of the paper) as a sensor. We used this sensor for testing the household dataset with the models proposed. As concerns the contributions of the papers, we modified the existing MCNN models adding from 3 to 6 convolutional layers to increase the performance of classification on the Fashion-MNIST dataset and we saw that our model boosted the state od the art works. Moreover, we tested the best MCCN model (the MCNN15) with other two dataset (the Fashion-Product dataset and a costumised dataset of ours to see how our model behaved with different datasets.
For what concerns the methodology, we described the basic MCNN and we modified the structure of the existing CNN to improve the accuracy and we compared our model performance with the state of art works and literature (see respectively table 2 and 3).
Please see the uploaded version of the paper in the attachments

Reviewer 2 Report
1- at the end of the introduction section, you should add a clear statement about the proposed method and the contribution of the paper.
2- the contribution of the paper is to apply MCNN to Fashion classification and create a relatively small dataset.
3- please add some figures for your custom fashion dataset
4- Table 3 shows two accuracies for the same model and one of them is better than the one proposed and there is no discussion about these results.
5- The custom dataset and the Fashion Product Images Dataset are very small how did you train and test the models?
it is a bit surprising that this old model MCNN is better than the state-of-the-art models such as efficientnet. why this happened?
Author Response
Response to Reviewer 2 Comments
Dear Editors and Reviewers, thank you for your letter and for the reviewers’ comments concerning our manuscript entitled “Image Classification using Multiple Convolutional Neural Networks on the Fashion-MNIST dataset”. We have analysed these valuable comments thoroughly and have made corrections which we think fully address the reviewers’ concerns, as described below.
Reviewer 2
Point 1: at the end of the introduction section, you should add a clear statement about the proposed method and the contribution of the paper.
Reply: We appreciate this comment. We added the proposed method with our contributions at the end of the introduction.
Point 2: the contribution of the paper is to apply MCNN to Fashion classification and create a relatively small dataset.
Reply: Thank you for your instructive suggestions. Following the reviewer’s comment we added in the introduction section the proposed method and the contributions of our work.
Point 3: please add some figures for your custom fashion dataset
Reply: I appreciate your great suggestion. We already prepared the figures for the household dataset and it it shown in Figure 3 (b).
Point 4:Table 3 shows two accuracies for the same model and one of them is better than the one proposed and there is no discussion about these results.
Reply: Thank you for your instructive suggestions. As concerns the two accuracies regarding the same model in Table 3, in the discussion section we mentioned that these accuracies are referred to the tool used (Tensorflow or Pytorch). In the discussion section it is stated that “In Table 3, the numbers in red represent the accuracy obtained using Pytorch. We realized that the model’s accuracy increased a lot when the authors trained their models
using Tensorflow2. Therefore, we replicated the models and trained them using PyTorch
again. As a result, accuracy is reduced as 90.64% compared to 98.8% on the CNN LeNet-5
model [16] and 90.25% compared to 99.1% on the CNNs model [16]”
Point 5:The custom dataset and the Fashion Product Images Dataset are very small how did you train and test the models? Reply: Thank you for your considerable comments. We used the Fashion Product and the costumised dataset only in the testing phase and we trained the models only using the Fashion-MNIST dataset. In subsection 3.1 we added this sentence to clarify this differentiation between the datasets used in the training phase and the datasets used during the testing part: “ We used the Fashion-MNIST dataset for training the models and we tested the architectures with the Fashion Product and the costumized dataset of ours.”
Point 6: it is a bit surprising that this old model MCNN is better than the state-of-the-art models such as efficientnet. why this happened?
Reply: Thank you for your considerable comments. We think that it was possible only to consider the Fashion-mnist dataset, which is small and consists of only one channel image. It is why we could propose a good performance network model, although our model is simple compared to efficientnet. Laster, we will apply for big datasets (e.g., imagenet dataset) that clearly confirm our model’s performance.
We think that our model boosted the state of the art models such as efficientnet thanks to the additional layers added in the end of the “standard” MCNN models. As [1] stated convolutional layers are also able to significantly reduce the complexity of the model through the optimisation of its output. Moreover, in [2] the authors stated that “MCNN has fewer parameters that a traditional CNN which makes the processing (in this case cloth classification) more efficient.”
References:
[1] O'Shea, Keiron, and Ryan Nash. "An introduction to convolutional neural networks." arXiv preprint arXiv:1511.08458 (2015).
[2] Xu, P.; Wang, L.; Guan, Z.; Zheng, X.; Chen, X.; Tang, Z.; Fang, D.; Gong, X.; Wang, Z. Evaluating brush movements for Chinese calligraphy: a computer vision based approach. In Proceedings of the 27th International Joint Conference on Artificial Intelligence, IJCAI 2018, 2018, pp. 1050–1056.
Please see the new version of the paper in attachments

Reviewer 3 Report
1. The main theme of this paper is not consistent. The abstract has mentioned about the demand of service robot in dressing and laundry activities for elderly population. However, this idea is rarely mentioned in the remaining of manuscript anymore. It is necessary authors to rewrite and restructure this manuscript to ensure the coherence of theme.
2. Affiliations of all authors are not provided.
3. Section 0 Article Highlights and Section 1 Introduction should be merged into a single section.
4. There are no problem statements mentioned in the Introduction section. It is not clear what kind of technical issues to be addressed in this manuscript. The overview of proposed work should be explained in this section as well.
5. The literature review should not only focus on the Fashion-MNIST dataset only and other relevant datasets should be covered as well.
6. Authors need to explain the differences between their proposed work and those surveyed in Section 2.
7. One of the main concerns on this paper is the novelty of multiple convolutional neural networks (MCNNs) model. It is not much different with typical CNN except more convolutional layers are added in MCNN. The technical contribution offered by MCNN is rather limited and it is not evident how MCNN can offer to the advancement of deep learning research. Authors need to provide further clarification on this issue and explain the additional values can be offered by MCNN.
8. What is the meaning of symbol "n" in Eq. (1). Is this same as "N" in Line 136?
9. Only two hyperparameters are mentioned in Table 1. How about other hyperparameters such as optimizer, batch size, epoch number, filter size, stride size and etc? Are they also optimized using Ray tune as well? If not, then authors need to justify how those hyperparameter settings in Lines 161 and 162 are obtained.
10. MCNN has shown good performance in terms of accuracy. But how about the network complexity in term of the number of parameters? Authors need to investigate the trade-off between accuracy and network complexity, especially when more convolutional layers are added.
11. Figure 4 - Authors are suggested to present the results for training and testing sets using different figures.
12. ROC curves and AUC values produced by the proposed work should be presented.
13. Please check the Abbreviation in Pg. 11. Looks like they are not arranged properly.
Author Response
Response to Reviewer 3 Comments
Dear Editors and Reviewers, thank you for your letter and for the reviewers’ comments concerning our manuscript entitled “Image Classification using Multiple Convolutional Neural Networks on the Fashion-MNIST dataset”. We have analysed these valuable comments thoroughly and have made corrections which we think fully address the reviewers’ concerns, as described below.
Reviewer 3
Point 1: The main theme of this paper is not consistent. The abstract has mentioned about the demand of service robot in dressing and laundry activities for elderly population. However, this idea is rarely mentioned in the remaining of manuscript anymore. It is necessary authors to rewrite and restructure this manuscript to ensure the coherence of theme.
Reply: We appreciate this comment. Following the reviewer’s suggestion we changed the first part of the abstract: “As the elderly population is growing there is a need for caregivers, which may become unsustainable for the society. In this scenario the demand for automated help grows. Service robotics is one area where robots have shown significant promise in working with people. Household settings, and elderly homes will need intelligent robots in use that perform daily activities. Cloth manipulation is one such daily activity and represents a challenging area for a robot and there is growing interest in studying dressing tasks that include detecting and classifying household objects”
Point 2: Affiliations of all authors are not provided. Reply: Thank you for your instructive suggestions. Following the reviewer’s comment we added the affiliations.
Point 3: Section 0 Article Highlights and Section 1 Introduction should be merged into a single section.
Reply: We appreciate your great suggestion. Following your comment, we merged the Article Highlights into the introduction section.
Point 4 There are no problem statements mentioned in the Introduction section. It is not clear what kind of technical issues to be addressed in this manuscript. The overview of proposed work should be explained in this section as well.
Reply: Thank you for your instructive suggestions. In order to explained the overview of the proposed work in the Introduction section, we added at the end of the introduction the contributions of our work. Moreover, we added the sentence “From the state of art it can be pointed out that using CNNs boost the accuracy of cloth image classification, so we decided to study in depth a specific case of CNNs, the multiple convolutional layers (MCNNs) since these models have fewer parameters than a traditional CNN which makes the networks more efficient [2] to pointed out the problem statements and the technical issues in the introduction section.
Point 5:The literature review should not only focus on the Fashion-MNIST dataset only and other relevant datasets should be covered as well.
Reply: Thank you for your considerable comments. We added this part in the section Related Work: “Many datasets are used for cloth image classification… of Indian ethnic clothes.” to have a full overview of the main datasets used for cloth image classification.
Point 6: Authors need to explain the differences between their proposed work and those surveyed in Section 2. Reply: Thank you for your considerable comments. The novelty of our work consisted on considering the MCNN to boost the accuracy of the cloth image classification with the Fashion-MNIST dataset. Moreover, to explain the differences between our method and those surveyed in section 2, we added the sentence “We chose to use this kind of network to boost the accuracy of cloth image classification since since these models have fewer parameters than a traditional CNN which makes the networks more efficient [2]”
Point 7: One of the main concerns on this paper is the novelty of multiple convolutional neural networks (MCNNs) model. It is not much different with typical CNN except more convolutional layers are added in MCNN. The technical contribution offered by MCNN is rather limited and it is not evident how MCNN can offer to the advancement of deep learning research. Authors need to provide further clarification on this issue and explain the additional values can be offered by MCNN.
Reply: Thank you for your considerable comments. In [2], the authors stated that “MCNN has fewer parameters than a traditional CNN which makes the processing (in this case cloth image classification) more efficient”. This is the main reason why we decided to use and modify this type of CNN.
Point 8: What is the meaning of symbol "n" in Eq. (1). Is this same as "N" in Line 136?.
Reply: Thank you for your considerable comments, the N was the n in capital letter since it was at the beginning of a sentence, but we change from N to n in the text.
Point 9: Only two hyperparameters are mentioned in Table 1. How about other hyperparameters such as optimizer, batch size, epoch number, filter size, stride size and etc? Are they also optimized using Ray tune as well? If not, then authors need to justify how those hyperparameter settings in Lines 161 and 162 are obtained. (Jaeseok)
Reply: Thank you for your considerable comments. You are right. We consider other hyperparameters described in Table 1. Also, we describe how those hyperparameter settings are in section 2.2.
Point 10:MCNN has shown good performance in terms of accuracy. But how about the network complexity in term of the number of parameters? Authors need to investigate the trade-off between accuracy and network complexity, especially when more convolutional layers are added.
Reply: Thank you for your considerable comments. Based on your advice, we explained the trade-off between accuracy and network complexity in section 5.
Point 11 Figure 4 - Authors are suggested to present the results for training and testing sets using different figures
Reply: Thank you for your considerable comments. We searched in other papers a different way to present our results but we found that in many works they usually presented the results using the accuracy and loss figures as we did.
Point 12:ROC curves and AUC values produced by the proposed work should be presented.
Reply: Thank you for your considerable comments. We added ROC curve graph and AUC values in section 4.
Point 13:Please check the Abbreviation in Pg. 11. Looks like they are not arranged properly.
Reply: Thank you for your considerable comments, we arranged properly the abbreviations in page 11.
References:
[1] O'Shea, Keiron, and Ryan Nash. "An introduction to convolutional neural networks." arXiv preprint arXiv:1511.08458 (2015).
[2] Xu, P.; Wang, L.; Guan, Z.; Zheng, X.; Chen, X.; Tang, Z.; Fang, D.; Gong, X.; Wang, Z. Evaluating brush movements for Chinese calligraphy: a computer vision based approach. In Proceedings of the 27th International Joint Conference on Artificial Intelligence, IJCAI 2018, 2018, pp. 1050–1056.

Round 2
Reviewer 1 Report
The authors have revised the paper based on the reviewer's comments. However, there are few things need to be improvised.
1) In Table 3, what does the value of bracket means? CNNs[23] 90.25% (99.1%) Please justify.
2) The caption of Table 4 and Table 5 should be on top of the table, not bottom.
Author Response
Response to Reviewer's Comments
Dear Editors and Reviewers, regarding our paper, "Image Classification using Multiple Convolutional Neural Networks on the Fashion-MNIST dataset," we appreciate your letter and the reviewers' remarks. Following these insightful remarks carefully, we have made the modifications below that, in our opinion, completely address the reviewers' concerns.
Reviewer 1
Point 1: In Table 3, what does the value of bracket means? CNNs[23] 90.25% (99.1%) Please justify.
Reply: Thank you for your instructive suggestions. As concerns the two accuracies regarding the same model in Table 3, in the discussion section, we mentioned that these accuracies referred to the tool used (Tensorflow or Pytorch). In the discussion section, it is stated that “In Table 3, the numbers in red represent the accuracy obtained using Pytorch. We realized that the model’s accuracy increased greatly when the authors trained their models
using Tensorflow2. Therefore, we replicated the models and trained them using PyTorch
again. As a result, accuracy is reduced to 90.64% compared to 98.8% on the CNN LeNet-5
model [16] and 90.25% compared to 99.1% on the CNNs model [16]”
Point 2: The caption of Table 4 and Table 5 should be on top of the table, not bottom.
Reply: We appreciate this comment. We modified the caption of Tables 4 and 5 on top of the tables.
Reviewer 2 Report
No further comments
Author Response
Response to Reviewer's Comments
Dear Editors and Reviewers, regarding our paper, "Image Classification using Multiple Convolutional Neural Networks on the Fashion-MNIST dataset," we appreciate your letter and the reviewers' remarks. Following these insightful remarks carefully, we have made the modifications below that, in our opinion, completely address the reviewers' concerns.
Reviewer 2: No further comments
Reply: Thank you for your previous comments.
Reviewer 3 Report
This paper has addressed most of my comment. However, there are still some areas that I think can be improved further as follows:
1. Linguistic issues such as grammatical error and typo can be found. Please proofread the manuscript again before the final submission.
2. Authors have claimed that the proposed MCNN have fewer parameters than a traditional CNN. But it is not clear which traditional CNNs are referred by the authors. Please provide the examples of these traditional CNNs and their number of parameters so that the motivations of current research can be more convincing.
3. The tradeoff between accuracy and network complexity should be presented in table format for better clarity purpose. Authors should also try to compare the network complexity of MCNN with those of existing methods so that the contributions of current research can be more convincing.
4. How the hyperparameters such as number of feature maps, number of hidden layers and etc. for MCNN variants in figure 1 are determined? Are these optimal hyparameters and determined using Ray tune? Authors need to provide further clarification within the text.
Author Response
Response to Reviewer's Comments
Dear Editors and Reviewers, regarding our paper, "Image Classification using Multiple Convolutional Neural Networks on the Fashion-MNIST dataset," we appreciate your letter and the reviewers' remarks. Following these insightful remarks carefully, we have made the modifications below that, in our opinion, completely address the reviewers' concerns.
Reviewer 3
Point 1: Linguistic issues such as grammatical errors and typos can be found. Please proofread the manuscript again before the final submission.
Reply: I appreciate your great suggestion. We proofread the manuscript again before the final submission.
Point 2: Authors have claimed that the proposed MCNN has fewer parameters than a traditional CNN. But it is not clear which traditional CNNs are referred by the authors. Please provide examples of these traditional CNNs and their number of parameters so that the motivations of current research can be more convincing.
Reply: We appreciate this comment. We added two examples of traditional CNNs, which are Alexnet and Resnet18, in the text (see lines 59 and 60).
Point 3: The tradeoff between accuracy and network complexity should be presented in table format for better clarity purposes. Authors should also try to compare the network complexity of MCNN with those of existing methods so that the contributions of current research can be more convincing.
Reply: Thank you for your valuable comments; it represents a good idea for our paper contribution. Based on your suggestion, we added a table and wrote a comparison of the network complexity with other existing methods in the Discussion section (lines 319-332).
Point 4: How the hyperparameters such as the number of feature maps, number of hidden layers and etc. for MCNN variants in figure 1 are determined? Are these optimal hyparameters and determined using Ray tune? Authors need to provide further clarification within the text.
Reply: We appreciate this comment. We decided on the hyperparameters based on our GPU memory performance. We wrote how to set the hyperparameters in the Methods section (lines 198-206).